# Determinants of Survival of HIV Patients Receiving Dolutegravir: A Prospective Cohort Study in Conflict-Affected Bunia, Democratic Republic of Congo

**DOI:** 10.3390/ijerph191610220

**Published:** 2022-08-17

**Authors:** Roger T. Buju, Pierre Z. Akilimali, Nguyen-Toan Tran, Erick N. Kamangu, Gauthier K. Mesia, Jean Marie N. Kayembe, Hippolyte N. Situakibanza

**Affiliations:** 1Department of Public Health, Faculty of Medicine, University of Bunia, Bunia P.O. Box 292, Congo; 2Department of Biostatistics and Epidemiology, Kinshasa School of Public Health, University of Kinshasa, Kinshasa P.O. Box 11850, Congo; 3Australian Centre for Public and Population Health Research, Faculty of Health, University of Technology Sydney, P.O. Box 123, Sydney, NSW 2007, Australia; 4Faculty of Medicine, University of Geneva, Rue Michel-Servet 1, 1206 Genève, Switzerland; 5Département des Sciences de Base, School of Medicine, University of Kinshasa, Kinshasa P.O. Box 11850, Congo; 6Unité de Pharmacologie Clinique et Pharmacovigilance, School of Medicine, University of Kinshasa, Kinshasa P.O. Box 11850, Congo; 7Department Internal Medicine, School of Medicine, University of Kinshasa, Kinshasa P.O. Box 11850, Congo; 8Department of Tropical Medicine, Infectious and Parasitic Diseases, School of Medicine, University of Kinshasa, Kinshasa P.O. Box 11850, Congo

**Keywords:** armed conflict, antiretroviral therapy, survival analysis, Bunia, Republic Democratic of Congo

## Abstract

This study aims to determine the factors influencing HIV-related mortality in settings experiencing continuous armed conflict atrocities. In such settings, people living with HIV (PLHIV), and the partners of those affected may encounter specific difficulties regarding adherence to antiretroviral therapy (ART), and retention in HIV prevention, treatment, and care programs. Between July 2019 and July 2021, we conducted an observational prospective cohort study of 468 PLHIV patients treated with Dolutegravir at all the ART facilities in Bunia. The probability of death being the primary outcome, as a function of time of inclusion in the cohort, was determined using Kaplan–Meier plots. We used the log-rank test to compare survival curves and Cox proportional hazard modeling to determine mortality predictors from the baseline to 31 July 2021 (endpoint). The total number of person-months (p-m) was 3435, with a death rate of 6.70 per 1000 p-m. Compared with the 35-year-old reference group, older patients had a higher mortality risk. ART-naïve participants at the time of enrollment had a higher mortality risk than those already using ART. Patients with a high baseline viral load (≥1000 copies/mL) had a higher mortality risk compared with the reference group (adjusted hazard ratio = 6.04; 95% CI: 1.78–20.43). One-fourth of deaths in the cohort were direct victims of armed conflict, with an estimated excess death of 35.6%. Improving baseline viral load monitoring, starting ART early in individuals with high baseline viral loads, the proper tailoring of ART regimens and optimizing long-term ART, and care to manage non-AIDS-related chronic complications are recommended actions to reduce mortality. Not least, fostering women’s inclusion, justice, peace, and security in conflict zones is critical in preventing premature deaths in the general population as well as among PLHIV.

## 1. Introduction

Globally, AIDS-related deaths have decreased since their peak in 2004, from 1.9 million in 2004 to 680,000 in 2021 [1]. As a result, the life expectancy of people living with HIV (PLHIV) has considerably improved due to highly effective antiretroviral (ARV), and the death rate among PLHIV is nearing that of the general population [2].

In the Democratic Republic of Congo (DRC), AIDS-related deaths decreased by 60% in 2018 [3]. However, this reduction in deaths will likely be reversed due to disrupted access to antiretroviral therapy (ART) by contextual factors, such as the COVID-19 pandemic, Ebola outbreaks, and continuous security threats.

Since 2017, armed conflict has negatively affected the provision of healthcare services in Bunia, the capital of the eastern province of Ituri, neighboring Uganda. Bunia is surrounded by violent combat zones, where the army and militia fight on a daily basis. Intermittently, clashes also occur in Bunia itself. Restricted access to healthcare services due to armed violence has been documented worldwide, such as in South America, where armed conflicts in Colombia have led to the forced displacement of millions of families [4]. In such insecure environments, the affected community may avoid visiting health facilities for fear of violence, rape, or even murder by the insurgents [5,6]. As a result, patients receiving ART may face challenges in following their routine ART programs and care schedules and experience a rebound in viral load (VL) and sometimes death [7]. Therefore, resistance to dolutegravir (DTG)-containing regimens as a result of poor adherence, may affect mortality [8].

Although the mortality rate among PLHIV has declined in the DRC, the chance of survival is not the same for all. Mortality predictors include advanced AIDS clinical stages, the presence of comorbidities and opportunistic infections, lower body mass index, gender, and high viral load during the first two years of illness [9,10,11,12,13]. In resource-limited and especially conflict-stricken settings, such as in the DRC, the factors contributing to the death of PLHIV on DTG-based regimens are not yet well-elucidated. Therefore, we conducted a prospective cohort study that aimed to investigate the survival determinants of PLHIV receiving DTG-containing ART in conflict-affected Bunia.

## 2. Methods

### 2.1. Study Design and Participants

Between July 2019 and July 2021, we conducted an observational prospective cohort study involving 468 adults living with HIV and receiving DTG-containing ART. Participants were recruited from all the health facilities offering ART in Bunia. At enrollment, we included ART-naïve participants as well as those already on ART.

### 2.2. Procedures, Data Collection, and Outcome

Upon enrollment, ART-naïve participants received a DTG-containing regimen (i.e., DTG 50 mg with lamivudine 300 mg and tenofovir disoproxil fumarate 300 mg, once daily). For participants already receiving ART, we switched their previous (DTG-free) treatment to the DTG-containing regimen. Previous DTG-free treatment included Stavudine (D4T) or Zidovudine (AZT), combined with Lamivudine (3TC) and either Nevirapine (NVP) or Efavirenz (EFV). Every six months, participants underwent a follow-up evaluation, which included a physical examination, a review of adverse events and concomitant medicines, as well as HIV-1 RNA VL, a hemogram, and liver, urine, and renal function tests [6]. As part of a standard follow-up, participants’ CD4 cell and complete blood counts were scheduled every three months [14]. Participants came to health facilities every month for their ARV refills.

ART facilities offered outreach follow-ups, which included home visits or phone calls by community health agents to participants or their families. Community health agents informed facility data clerks after every outreach visit to participants. After three unsuccessful home visits, community health agents called participants up to five times. We deemed participants lost to follow-up (LTFU) if they were not seen at their designated ART facility for at least 90 days following the previous missed visit and had not transferred to another ART facility. The LTFU date was defined as the day of the last visit to the facility according to the medical records. If a family member, neighbor, or community leader reported the participant’s death, we categorized the status of that participant as dead [14].

We gathered sociodemographic data (age, sex, marital status, and education), clinical data (status of treatment before enrollment, tobacco use, alcohol use, WHO HIV clinical stage, and months of exposure to DTG-based regimen), and biological data (serum creatinine at baseline, hemoglobin, alanine transaminase (ALAT), syphilis serology, and hepatitis B serology (AgHBs)). We divided participants into three groups according to their treatment status before enrollment: (1) patients with ART experience before enrollment, (2) self-reported ART-naïve patients with baseline VLs of <50 copies/mL, and (3) self-reported ART-naïve patients with baseline VLs of ≥50 copies/mL [6]. Based on WHO criteria, anemia was defined as a hemoglobin concentration of <12 g/dL for women and <13 g/dL for men [15].

Death was the primary outcome. We investigated the cause of death in every case, notably deaths after fighting or ambush episodes.

### 2.3. Statistical Analysis

We recorded data using Epi Info 7 Version 7.2.0.1. software (Centers for Disease Control and Prevention, Atlanta-Georgia, GA, USA). We checked the data for completeness before entry and performed data double entry.

We calculated proportions, where the main outcome variable was death. We determined the incidence rate of recorded death events per 1000 person-months (p-m) from the date of enrollment. For participants known to have transferred out or LTFU, we censored the data at the date of the last appointment. For participants still in active care at the end of the study period, data was censored on the date of their last clinic visit. We used Kaplan–Meier curves to determine the probability of death as a function of the time of inclusion in the cohort, the log-rank test to compare survival curves based on determinants, and Cox proportional-hazard modeling to measure predictors of death from enrollment to the endpoint, which was set on 31 July 2021. We included the following variables in the Cox regression model: sex, age, WHO HIV clinical stage, baseline serum creatinine, baseline viral load, and treatment status before enrollment.

The proportionality test based on Schoenfeld residuals verified compliance with the assumption of the proportionality of risks. We assessed multicollinearity using variance inflation factors (VIFs) greater than 4.0. To identify the profile of patients who died because of armed conflict, we reported their causes of death according to six potential determinants (age, sex, WHO HIV clinical stage, baseline serum creatinine, baseline VL, and treatment status before enrollment). The proportionality test based on Schoenfeld residuals verified compliance with the assumption of the proportionality of risks. All the tests were two-sided, with the level of significance was set as *p* < 0.05. We performed all tests using Stata software version 14 (StataCorp, College Station, TX, USA).

### 2.4. Ethical Statement

The Kinshasa University School of Public Health’s institutional review board for research subjects approved the study protocol (no. app.: ESP/CE/094/2018 of August 2018). Before participating in the study, all participants signed a written informed consent form. Before analysis, all data were anonymized and deidentified [6].

## 3. Results

### 3.1. Patient Characteristics at Enrollment and Follow-Up Status

Of the 468 enrolled participants, 69.4% were women, 65.6% had no education or only primary education, and 56% lived alone. The average age of the participants was 38.97 years (SD: 11.94). Other characteristics are presented in Table 1.

At baseline, 219 individuals (46.8%) were diagnosed with WHO HIV stage I or II. A third of the participants (33.8%) had an abnormal blood creatinine level, and one out of four (27.6%) was anemic. Regarding baseline treatment status, 62.2% were already receiving ART (experienced patients), 19.9% self-reported as naïve with a baseline VL of <50 copies/mL (new but with VL suppressed), and 17.9% self-reported as naïve with a baseline VL of ≥50 copies/mL (new but with high VL) (Table 2).

### 3.2. LTFU and Mortality Rate

During the study period, more than a quarter of the cohort was LTFU (28.8%; 95% CI: 24.9–33.1). The overall number of person-months (p-m) followed up was 3435.22, with 33.48 LTFU per 1000 p-m. After 1, 3, 6, 9, and 12 months, 12.0%, 21.4%, 26.5%, 28.6%, and 28.8% were lost to follow-up, respectively.

The mortality of PLHIV on ART after 12 months was 4.9% (95% CI: 3.3–7.3), with an overall incidence rate of 6.70 deaths per 1000 p-m.

### 3.3. Predictors of Death among Patients with HIV on ART

After adjustment, older participants had a higher risk of death than the reference group (<35 years) (adjusted hazard ratio (aHR) for the 35–45 age group: 5.14; 95% CI: 1.09–24.18; aHR for the >45 age group: 5.47; 95% CI: 1.13–26.38). ART-naïve participants with a baseline VL of <50 copies/mL (i.e., suppressed) had a higher risk of death than participants already receiving ART (aHR: 5.43; 95% CI: 1.59–18.48). Participants with a high baseline VL (≥1000 copies/mL) had a higher mortality risk than the reference group (aHR: 6.04; 95% CI: 1.78–20.43) (Table 3 and Figure 1, Figure 2 and Figure 3).

Among the 23 reported deaths, 6 (26.1%) were directly related to armed conflict. Although not statistically significant, the proportion of deaths as a direct result of armed conflict was higher among women and among those who were at stage I or II. While statistically insignificant, patients who died from armed conflict were younger than their counterparts (Table 4). Excluding the six deaths due to armed conflict, the death rate was 4.94 per 1000 person-months. The mean VIF was 1.73, which we found to be insignificant.

## 4. Discussion

In this prospective study, we analyzed the pattern of mortality in a cohort of 468 PLHIV receiving DTG-containing ART from health facilities located in the conflict-affected city of Bunia, eastern DRC. The factors associated with increased mortality were being aged 35 and above, having no ART before study enrollment, and high baseline VL. The death rate was 6.70 per 1000 person-months. One in four deaths was due to armed conflict.

Armed conflict can affect cohort mortality rates directly or indirectly: directly if PLHIV fall victim to hostilities, or indirectly if the population’s forced displacement or fear of movement restricts their access to HIV treatment and care [16]. The death rate of 6.70 per 1000 person-months in our study included fatalities related (6 cases) and unrelated (17 cases) to armed conflict. The baseline characteristics did not differ significantly between the two groups. This suggests that the cause of death was likely not to have been related to differences in age, gender, and biological factors, suggesting that a restricted access to HIV treatment and care due to armed conflict may have affected both groups in similar ways. What about the significance of death directly caused by armed conflict? Based on the death rate of 4.94 per 1000 person-months, if there were no fatalities due to armed conflict, the excess death due to armed conflict is estimated to be 35.6% [(6.70 − 4.94)/4.94] × 100. However, this figure is not reliable as the difference between the two proportions (6.70/1000 and 4.94/1000) is not statistically significant (*p* = 0.429). A larger sample size of at least 1000 would be necessary to detect a statistically significant difference with a two-tailed *p*-value ≤ 0.05 and to determine whether armed conflict plays a significant role in raising the mortality rate among PLHIV.

The findings correspond to those of previous research in low-income countries, which also found that mortality is associated with patient age, baseline VL, and the status of treatment before enrollment [17,18]. The mortality reported in our study is within the ranges reported in a systematic review and meta-analysis, where the mortality rate per 1000 person-months was 9.0 (95% CI 5.8–12.2%) at 12 months of follow-up [17]. Another study conducted in the DRC, in the conflict setting of Goma, found a death rate of 7.3% at 12 months of follow-up [9].

As previously documented in Kenya, Mozambique, Rwanda, and Tanzania, the risk of death is significantly higher in older than younger PLHIV enrolled in ART programs [19]. In a study from South Africa, when compared with younger patients, older patients experienced a worse immunologic recovery on ART, as evidenced by a lower CD4+ cell count recovery [20]. When compared with HIV-negative peers of the same age, research found that comorbidities tend to arise earlier in older HIV patients [21]. The older cohort had a ten-fold higher prevalence of hypertension and diabetes than the younger cohort. Comorbidities, such as hypertension and diabetes, were more common in the older group, which might indicate a higher likelihood of contact with healthcare services and, as a result, an earlier start of ART in accordance with HIV treatment guidelines. However, ARVs must be adjusted to have the least negative impact on other comorbidities [21].

In our study, ART-naïve patients had a higher risk of death compared with patients already receiving ART at baseline. Increased treatment experience among patients already receiving ART and the resulting VL suppression at baseline could explain their lower rate of death compared with ART-naïve patients. In the study undertaken in Goma, the likelihood of non-suppression remained significantly lower among ART-naïve patients with a baseline VL of ≥50 copies/mL compared with patients already receiving ART before being included in the study [9].

In terms of policy and practice implications stemming from our research in Bunia, we recommend strengthening baseline VL measurement for all PLHIV. Currently, routine baseline VL testing for all PLHIV is not recommended and cannot be undertaken in all healthcare facilities in Bunia (VL tests are sent to Kinshasa). Efforts should be devoted toward increasing the availability of low-cost equipment for VL measurement. Early ART initiation among patients with a high baseline VL, and optimizing long-term ART and care to also manage non-AIDS-related chronic complications, could help limit mortality [22,23]. Most importantly, fostering women’s inclusion, justice, peace, and security in conflict zones is critical in preventing premature deaths among PLHIV as well as the general population [24].

This study has several limitations, the first being that, due to insecurity, we could not always fully follow up and ascertain the real outcomes of the participants who were documented as LTFU (we report a higher LTFU compared with previous studies [18,25]. Second, like any cohort analysis with right-censored observations, the mortality rate may be skewed downward if unreported deaths are categorized as LTFU. Third, our sampling size was too small to confidently determine the association between armed conflict and death among PLHIV. However, even one death due to armed conflict is an unnecessary death—a death in excess. Fourth, as mentioned above, armed conflict can affect cohort mortality rates directly (death due to fighting) or indirectly (restricted access to HIV treatment and care) [26]. We did not investigate the latter indirect relationship, nor did we investigate other potential mortality predictors, such as adherence to ART, malnutrition, immunologic and virologic failure, and comorbidities (e.g., diabetes, high blood pressure, and other infections). Nevertheless, to our knowledge, our study is among the limited research quantitatively and prospectively assessing mortality among PLHIV in conflict settings. These results can help researchers, healthcare workers, and other stakeholders involved in HIV treatment and care to understand the incidence and determinants of death in conflict-affected settings.

## 5. Conclusions

One-fourth of deaths in the PLHIV cohort was directly due to armed conflict, with an estimated excess death rate of 35.6%. To reduce mortality, we recommend strengthening baseline VL measurement among all PLHIV, initiating ART early in PLHIV with high VL, and optimizing ART and care to also manage non-AIDS-related chronic complications. Not least, fostering women’s inclusion, justice, peace, and security in conflict zones is critical in preventing premature deaths in the general population as well as among PLHIV.

## Figures and Tables

**Figure 1 ijerph-19-10220-f001:**
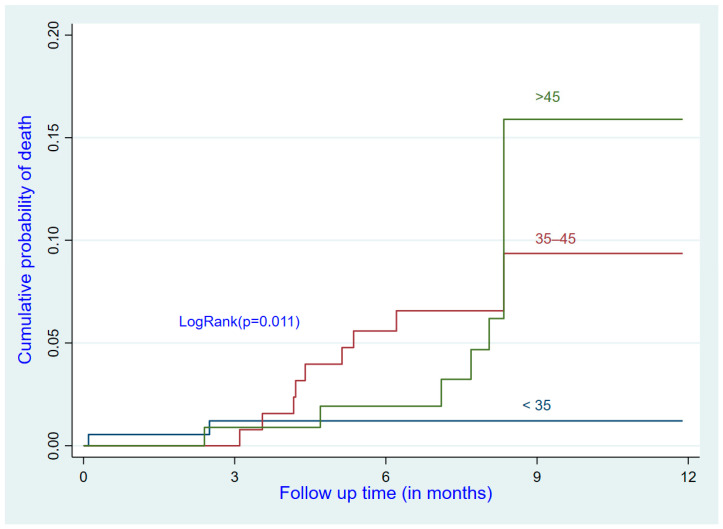
Cumulative incidence of death by age group.

**Figure 2 ijerph-19-10220-f002:**
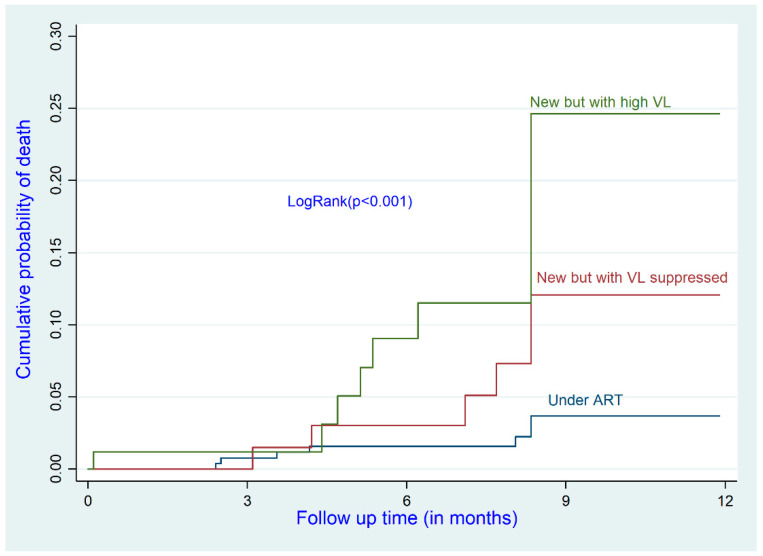
Cumulative incidence of death by status of treatment before enrollment.

**Figure 3 ijerph-19-10220-f003:**
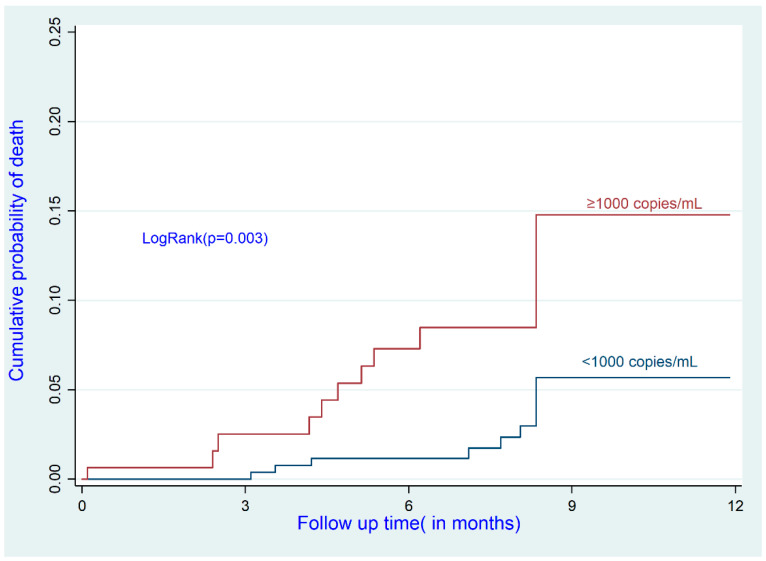
Cumulative incidence of death by viral load at baseline.

**Table 1 ijerph-19-10220-t001:** Baseline sociodemographic characteristics (*n* = 468).

	*n*	%
Mean age (SD)	38.97 (11.94)	
Age (years)		
<35	183	39.1
35–45	161	34.4
>45	124	25.5
Sex		
Female	325	69.4
Male	143	30.6
Education		
None/primary	307	65.6
Secondary/tertiary	161	34.4
Marital status		
Living alone	263	56.2
In union	205	43.8
Ethnic group		
Nilotic	224	47.9
Bantou	114	24.4
Semi-Bantou	93	19.9
Sudanese	37	7.9
Tobacco consumption		
No	357	76.3
Yes	111	23.7
Alcohol consumption		
No	265	56.6
Yes	203	43.4
Total	468	100.0

**Table 2 ijerph-19-10220-t002:** Clinical and biological baseline characteristics (*n* = 468).

Baseline Characteristics	*n*	%
HIV stage		
I and II	219	46.8
III and IV	249	53.2
Status of treatment at baseline		
Receiving ART	291	62.2
New but with VL suppressed	93	19.9
New but with high VL	84	17.9
Viral load at baseline		
<1000 copies/mL	313	66.9
≥1000 copies/mL	155	33.1
AgHBs		
Negative	449	95.9
Positive	19	4.1
RPR		
Negative	397	84.8
Positive	71	15.2
Hemoglobin mean (SD)	13.84 (2.77)	
Anemia		
No	339	72.4
Yes	129	27.6
Creatinine		
Normal	310	66.2
Abnormal	158	33.8
ALAT		
Normal	431	92.1
Abnormal	37	7.9
Loss to follow-up		
No	333	71.2
Yes	135	28.8
Deceased		
No	445	95.1
Yes	23	4.9
Total	468	100.0

**Table 3 ijerph-19-10220-t003:** Multivariate analysis of predictors of mortality (*n* = 468).

	*n*	Events (Death)	Person-Months	Incidence Rate of Death	Adjusted HR	95%CI	*p*-Value ^(1)^
Age (years)							
<35	183	2	1310.9	1.53	1		
35–45	161	10	1177.23	8.49	5.14	1.09–24.18	0.038
>45	124	11	947.09	11.61	5.47	1.13–26.38	0.034
Gender							
Female	325	11	2428.58	4.53	1		
Male	143	12	1006.64	11.92	1.27	0.50–3.18	0.610
WHO HIV stage							
Stage I and II	219	8	1795	4.46	1		
Stage III and IV	249	15	1640.22	9.15	1.38	0.57–3.35	0.469
Creatinine							
Normal	310	12	2421.23	4.96	1		
Abnormal	158	11	1013.99	10.85	1.84	0.79–4.25	0.153
Status of treatment before enrollment							
Under ART	291	7	2288.08	3.06	1		
Naïve but suppressed VL	93	6	676.68	8.87	5.43	1.59–18.48	0.007
Naïve but high VL	84	10	470.46	21.26	2.29	0.78–6.74	0.131
Viral load at baseline							
<1000 copies/mL	313	10	2458.52	4.07	1		
≥1000 copies/mL	155	13	976.7	13.31	6.04	1.78–20.43	0.004
Overall	468	23	3435.22	6.70			

HR: hazard ratio. ^(1)^: from Cox regression.

**Table 4 ijerph-19-10220-t004:** Distribution of causes of death (*n* = 23).

Baseline Characteristics	Related to Armed Conflict				*p*-Value *
	Yes		No		
	*n*	%	*n*	%	
Age, mean	44.83 ± 15.39		47.76 ± 12.27		ns
Age (years)					ns
<35	1	16.7	1	5.9	
35–45	1	16.7	9	52.9	
>45	4	66.6	7	41.2	
Sex					ns
Female	4	66.7	7	41.2	
Male	2	33.3	10	58.8	
WHO HIV stage					ns
Stage I or II	4	66.7	4	23.5	
Stage III or IV	2	33.3	13	76.5	
Creatinine					ns
Normal	2	33.3	10	58.8	
Abnormal	4	66.7	7	41.2	
Status of treatment before enrollment					ns
Receiving ART	2	33.3	5	29.4	
New but with VL suppressed	2	33.3	4	23.5	
New but with high VL	2	33.3	8	47.1	
Viral load at baseline					ns
<1000 copies/mL	3	50.0	7	41.2	
≥1000 copies/mL	3	50.0	10	58.8	
Total	6	26.1	17	73.9	

* Mann–Whitney test or Fisher’s test. ns: not significant.

## Data Availability

Data are available upon request from pierretulanefp@gmail.com.

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
