# Peer review of "Determinants of Survival of HIV Patients Receiving Dolutegravir: A Prospective Cohort Study in Conflict-Affected Bunia, Democratic Republic of Congo"

_ijerph, 2022, doi:10.3390/ijerph191610220_

Round 1
Reviewer 1 Report
This is a follow up review of the paper entitled, 'Determinants of Survival of HIV Patients Receiving Dolutegravir: A Prospective Cohort Study in Conflict-Affected Bunia, Democratic Republic of Congo'. The hypothesized that that armed conflicts in Bunia and the surrounding areas have significant impact on mortality among people living with HIV. I pointed out in my previous review that the data presented do not support that hypothesis. Therefore, the authors toned down (even remover) sections regarding war conflicts association with mortality. in the discussion, the authors mentioned that "unfortunately, we cannot analyze the relationship between armed conflicts and death among PLHIV due to small number".
I do not agree with this statement for two reasons. First, the authors did analyze the effect of war conflict. This was mentioned in the previous draft. Indeed, the purpose of this study was to determine the effect of war on mortality among PLHIV. Second, the authors mention the small number made it hard to analyze. The sample size was 468. If this is two small, there is no mention about the power calculation that made them to decide that their study population was too small to evaluate the effect of war conflict on mortality. Indeed, this is center to the reason why this study was conducted. Third, lack of disclosing a negative result. In this version, the authors removed areas where they previously discussed the effect of armed conflict. Instead of removing completely the effect of war on mortality, the authors must at least say that war did not play a significant role in raising the mortality rate significantly. Not mentioning this fact would be a failure to disclose a negative finding.
Minor problems:
Introduction: ... we categorized the status as dead after extensive follow-up'. Remove 'after extensive follow-up'.
Introduction: 3rd paragraph. Change the survival chance to chance or survival.
Paragraph 3: Male sex, should be changed to gender.
Result section:17.9% were self-reported naive. Change to 17.9% self-reported as naive with..."
Author Response
Reviewer#1
This is a follow up review of the paper entitled, 'Determinants of Survival of HIV Patients Receiving Dolutegravir: A Prospective Cohort Study in Conflict-Affected Bunia, Democratic Republic of Congo'. The hypothesized that that armed conflicts in Bunia and the surrounding areas have significant impact on mortality among people living with HIV. I pointed out in my previous review that the data presented do not support that hypothesis. Therefore, the authors toned down (even remover) sections regarding war conflicts association with mortality. in the discussion, the authors mentioned that "unfortunately, we cannot analyze the relationship between armed conflicts and death among PLHIV due to small number".
I do not agree with this statement for two reasons. First, the authors did analyze the effect of war conflict. This was mentioned in the previous draft. Indeed, the purpose of this study was to determine the effect of war on mortality among PLHIV. Second, the authors mention the small number made it hard to analyze. The sample size was 468. If this is two small, there is no mention about the power calculation that made them to decide that their study population was too small to evaluate the effect of war conflict on mortality. Indeed, this is center to the reason why this study was conducted. Third, lack of disclosing a negative result. In this version, the authors removed areas where they previously discussed the effect of armed conflict. Instead of removing completely the effect of war on mortality, the authors must at least say that war did not play a significant role in raising the mortality rate significantly. Not mentioning this fact would be a failure to disclose a negative finding.
Authors’ response: Many thanks for helping us reenter our perspectives on the impact of armed conflict. We have reintegrated the contents on this subject in the manuscript, offering additional discussion on the subject.
Minor problems:
Introduction: ... we categorized the status as dead after extensive follow-up'. Remove 'after extensive follow-up'.
Introduction: 3rd paragraph. Change the survival chance to chance or survival.
Paragraph 3: Male sex, should be changed to gender.
Result section:17.9% were self-reported naive. Change to 17.9% self-reported as naive with..."
Authors’ response: Many thanks for spotting these minor issues. We have now made the necessary changes.

Reviewer 2 Report
I cannot see the original comments, but I am going to assume that all my suggestions have been applied.
Author Response
Thanks
This manuscript is a resubmission of an earlier submission. The following is a list of the peer review reports and author responses from that submission.
Round 1
Reviewer 1 Report
I was enthused reading a study such as this done in an active war zone. The authors must be commended for pulling off a study in that context. The authors describe the results of a study entitled, "Determinants of Survival of HIV Patients Receiving Dolutegravir in Bunia, Democratic Republic of Congo: A Prospective Cohort Study", showing the factors associated with mortality in patients being treated with Dolutegravir.Overall. the study was designed and conducted appropriately. However, my enthusiasm was dumped by the presentation. The paper needs to be re-written and proofread by a native english speaker or someone with the command of the English language.
Examples are too numerous to list them all. Here is a list of them:
1. "However, in resource-limited countries, such as the Democratic Republic of Congo (DRC), this reduction in deaths will likely be reversed due to the disruption of access to ARV, caused by multiple health problems including the COVID-19 pandemic, Ebola, and insecurity, including pre-existing resistance to nucleoside reverse-transcriptase inhibitors coadministered with dolutegravir (DTG) [3,4]. Although the number of AIDS-related deaths decreased by 60% in 2018 in the DRC, the difficult of accessing ARV in the eastern region as a result of armed conflicts indicates that deaths will again increase." This paragraph is confusing, or simply contradictory.
2. "Particularly since 2017, the city of Bunia, located in the eastern part of the DRC in the vicinity of Uganda, has suffered the negative effects of armed conflicts that negatively affect the provision of health care."
Vicinity is not the correct word. Bunia is located at about 342 KM to Kasenyi, the closet Ugandan town.
3. "These types of events can prevent patients from attending health facilities because of fear of threats, rapes, or even murder by the insurgents in areas where insecurity reigns [6, 7]." This sentence could be improved. What do they mean by "Fear of threat"?
4. "Study Design and Participants: We performed an observational prospective cohort study of 468 HIV patients receiving DTG in all of Bunia's health institutions between July 2019 and July 2021. Bunia is a city in the DRC's eastern region. Since 2017, the city has been engulfed in violent warfare, which in certain sections continues to this day. Patients aged 18 and above were enrolled in this research (details are available elsewhere) [6]. Pregnant women were excluded. All the clinics were in Bunia, which is surrounded by violent combat zones where regular army and militia soldiers battle on a daily basis. As previously described, clashes occur from time to time in the city of Bunia. [6].
This paragraph needs to be re-written.
"Certain section". Is it within the City or in the surrounding region?
5. "Upon enrollment, we switched participants to DTG 50 mg with lamivudine 300 mg and tenofovir disoproxil fumarate 300 mg once daily."
This sentence is vague. Switched from what?
6. A questionnaire was developed to collect sociological and ethnic data from the participants. The data was collected by trained interviewers from the study team. Following the pre-testing with the first ten patients, we reviewed the questions for consistency and made any required changes. Data collection was overseen by designated supervisors to guarantee data validity and consistency, as well as adherence with ethical requirements (published in detail elsewhere) [6].
Vague, this could be improved.
6. When a patient was not seen at the ART clinic for at least 90 days or 3 months following the previous missed visit but had not transferred out of the facility to another facility, we deemed them as loss to follow-up (LTFU). For tracking patients' LTFU, we devised a unique approach. The health facilities offered extended follow-up, which included home visits or phone calls by community health agents. Patients or their families were contacted via phone using information gathered during health facility enrollment and home visits. The community health agents had completed high school and had received additional HIV/AIDS training. They informed the data clerks of each health facility about each patient's status after every visit. After three home visits from community health agents, each patient received five follow-up calls. If a family member, neighbor, or community leader reported the patient's death, we categorized the patient's status as dead after an extensive follow-up [14].
Poor style and confusing.
7) Definition of full-blown AIDS is not Stage I and Stage II: "At baseline, 219 individuals (46.8%) were diagnosed with full-blown AIDS (WHO stage I or II)."
8) Referencing the article [6] is often poorly presented. For example, "... as defined in the previous article [6]". Authors should have briefly described rather than leaving important information for the readers to go to a paper they published earlier this year.
9) Table 4: it is not clear whether armed conflict played any roles in mortality.
10) Discussion: In this study, we analyzed the pattern of mortality among a cohort of 468 at-risk patients with HIV in Bunia, DRC, over 3435.22 person-months, and identified determinants of mortality among patients on a DTG-based regimen.
At-risk of what?
11) etc.
Author Response
Comments and Suggestions for Authors
I was enthused reading a study such as this done in an active war zone. The authors must be commended for pulling off a study in that context. The authors describe the results of a study entitled, "Determinants of Survival of HIV Patients Receiving Dolutegravir in Bunia, Democratic Republic of Congo: A Prospective Cohort Study", showing the factors associated with mortality in patients being treated with Dolutegravir.
Overall. the study was designed and conducted appropriately. However, my enthusiasm was dumped by the presentation. The paper needs to be re-written and proofread by a native english speaker or someone with the command of the English language.
Authors’ response: thank you for your feedback, which has helped us improve our manuscript. We have thoroughly reviewed the paper.
Examples are too numerous to list them all. Here is a list of them:
- "However, in resource-limited countries, such as the Democratic Republic of Congo (DRC), this reduction in deaths will likely be reversed due to the disruption of access to ARV, caused by multiple health problems including the COVID-19 pandemic, Ebola, and insecurity, including pre-existing resistance to nucleoside reverse-transcriptase inhibitors coadministered with dolutegravir (DTG) [3,4]. Although the number of AIDS-related deaths decreased by 60% in 2018 in the DRC, the difficult of accessing ARV in the eastern region as a result of armed conflicts indicates that deaths will again increase." This paragraph is confusing, or simply contradictory.
Authors’ response: we have clarified the text, which now reads as follows:
“In the Democratic Republic of Congo (DRC), AIDS-related deaths decreased by 60% in 2018 [3]. However, this reduction in deaths will likely be reversed due to disrupted access to antiretroviral therapy (ART) by contextual factors, such as the COVID-19 pandemic, Ebola outbreaks, and continuous insecurity threats. At the patient level, re-sistance to dolutegravir (DTG)-containing regimens may also affect mortality [4].”
- "Particularly since 2017, the city of Bunia, located in the eastern part of the DRC in the vicinity of Uganda, has suffered the negative effects of armed conflicts that negatively affect the provision of health care."
Vicinity is not the correct word. Bunia is located at about 342 KM to Kasenyi, the closet Ugandan town.
Authors’ response: we have clarified the text, which now reads as follows:
“Particularly since 2017, armed conflicts have negatively affected the provision of healthcare services in Bunia, the capital of the eastern province of Ituri, neighboring Uganda. Bunia is surrounded by violent combat zones, where the army and militia fight on a daily basis. Intermittently, clashes also occur in Bunia itself. Restricted access to healthcare services due to armed violence has been documented worldwide, such as in South America, where armed conflicts in Colombia have led to the forced displacement of millions of families [5].”
- "These types of events can prevent patients from attending health facilities because of fear of threats, rapes, or even murder by the insurgents in areas where insecurity reigns [6, 7]." This sentence could be improved. What do they mean by "Fear of threat"?
Authors’ response: we have clarified the text, which now reads as follows:
“In such insecure environments, the affected community may avoid visiting health facilities for fear of violence, rape, or even murder by the insur-gents [6, 7]. As a result, patients on ART may face challenges in following their routine ART programs and care schedules and experience a rebound in viral load (VL) and sometimes death [8].”
- "Study Design and Participants: We performed an observational prospective cohort study of 468 HIV patients receiving DTG in all of Bunia's health institutions between July 2019 and July 2021. Bunia is a city in the DRC's eastern region. Since 2017, the city has been engulfed in violent warfare, which in certain sections continues to this day. Patients aged 18 and above were enrolled in this research (details are available elsewhere) [6]. Pregnant women were excluded. All the clinics were in Bunia, which is surrounded by violent combat zones where regular army and militia soldiers battle on a daily basis. As previously described, clashes occur from time to time in the city of Bunia. [6].
This paragraph needs to be re-written.
"Certain section". Is it within the City or in the surrounding region?
Authors’ response: we have clarified the text, which now reads as follows:
“2.1. Study Design and Participants
Between July 2019 and July 2021, we conducted an observational prospective cohort study involving 468 adults living with HIV and receiving DTG-containing ART. Partic-ipants were recruited from all the health facilities offering ART in Bunia. At enrollment, we included ART-naïve participants as well as those already on ART.”
- "Upon enrollment, we switched participants to DTG 50 mg with lamivudine 300 mg and tenofovir disoproxil fumarate 300 mg once daily."
This sentence is vague. Switched from what?
Authors’ response: we have clarified the text, which now reads as follows:
2.2. Procedures, Data Collection, and Outcome
“Upon enrollment, ART-naïve participants received a DTG-containing regimen (i.e., DTG 50 mg with lamivudine 300 mg and tenofovir disoproxil fumarate 300 mg, once daily). For participants already on ART, we switched their previous (DTG-free) treat-ment to the DTG-containing regimen. Previous DTG-free treatment included Stavudine (D4T) or Zidovudine (AZT), combined with Lamivudine (3TC) and either Nevirapine (NVP) or Efavirenz (EFV). Every six months, participants underwent a follow-up eval-uation, which included a physical examination, review of adverse events and concom-itant medicines, as well as HIV-1 RNA VL, a hemogram, and liver, urine, and renal function tests [6]. As part of standard follow-up, participants' CD4 cell and complete blood counts were scheduled every three months [14]. Participants came to health fa-cilities every month for their ARV refills.
ART facilities offered outreach follow-ups, which included home visits or phone calls to participants or their families by community health agents. Community health agents informed facility data clerks after every outreach visit to participants. After three unsuccessful home visits, community health agents called participants up to five times. We deemed participants lost to follow-up (LTFU) when they were not seen at their designated ART facility for at least 90 days following the previous missed visit and had not transferred to another ART facility. The LTFU date was defined as the day of the last visit to the facility according to the medical records. If a family member, neighbor, or community leader reported the participant's death, we categorized the status as dead after an extensive follow-up [14].”
- A questionnaire was developed to collect sociological and ethnic data from the participants. The data was collected by trained interviewers from the study team. Following the pre-testing with the first ten patients, we reviewed the questions for consistency and made any required changes. Data collection was overseen by designated supervisors to guarantee data validity and consistency, as well as adherence with ethical requirements (published in detail elsewhere) [6].
Vague, this could be improved.
Authors’ response: we have reworked this section as detailed in the response to the above questions.
- When a patient was not seen at the ART clinic for at least 90 days or 3 months following the previous missed visit but had not transferred out of the facility to another facility, we deemed them as loss to follow-up (LTFU). For tracking patients' LTFU, we devised a unique approach. The health facilities offered extended follow-up, which included home visits or phone calls by community health agents. Patients or their families were contacted via phone using information gathered during health facility enrollment and home visits. The community health agents had completed high school and had received additional HIV/AIDS training. They informed the data clerks of each health facility about each patient's status after every visit. After three home visits from community health agents, each patient received five follow-up calls. If a family member, neighbor, or community leader reported the patient's death, we categorized the patient's status as dead after an extensive follow-up [14].
Poor style and confusing.
Authors’ response: we have reworked this section as detailed in the response to the above questions.
7) Definition of full-blown AIDS is not Stage I and Stage II: "At baseline, 219 individuals (46.8%) were diagnosed with full-blown AIDS (WHO stage I or II)."
Authors’ response: we have clarified the text, which now reads as follows:
“At baseline, 219 individuals (46.8%) were diagnosed with WHO HIV stage I or II.”
8) Referencing the article [6] is often poorly presented. For example, "... as defined in the previous article [6]". Authors should have briefly described rather than leaving important information for the readers to go to a paper they published earlier this year.
Authors’ response: we have removed this type of referencing across the manuscript and described the related information as relevant.
9) Table 4: it is not clear whether armed conflict played any roles in mortality.
Authors’ response: we have clarified the text explaining Table 4, which now reads as follows:
“Among the 23 reported deaths, six (26.1%) were directly related to armed conflicts. Although not statistically significant, the proportion of deaths as a direct result of armed conflict was higher among women and among those who were at stage I or II. Patients who died directly from armed conflict were younger than their counterparts (Table 4).”
10) Discussion: In this study, we analyzed the pattern of mortality among a cohort of 468 at-risk patients with HIV in Bunia, DRC, over 3435.22 person-months, and identified determinants of mortality among patients on a DTG-based regimen.
At-risk of what?
Authors’ response: we have clarified the text and reworked the discussion, which now reads as follows:
“In this prospective study, we analyzed the pattern of mortality in a cohort of 468 PLHIV receiving DTG-containing ART from health facilities located in the conflict-affected city of Bunia, eastern DRC. The factors associated with increased mortality were age 35 and above, no ART before study enrollment, and high baseline VL. The death rate was 6.70 per 1000 person-months. One in four deaths was due to armed conflicts.”
11) etc.
Authors’ response: we have thoroughly reworked the manuscript from the abstract to the conclusion. Many thanks once again for your comments.
Reviewer 2 Report
Review of Determinants of Survival of HIV Patients Receiving Dolutegravir in Bunia, Democratic Republic of Congo: A Prospective Cohort Study
This is a well done study of HIV outcomes in a city near a region of conflict in the DRC. The authors have clearly stated they research goals, their data collection procedures and analytic methods. They have presented their results in a concise and clear manner. This article will be of interest to researchers to those working on HIV related topics and conflict/health. I do not feel as there is much that should prevent publication of this manuscript in IJERPH, but have the following suggestions.
Major issues:
The authors make many statements in the introduction without providing evidence or supporting information.
Example:
“Particularly since 2017, the city of Bunia, located in the eastern part of the DRC in the vicinity of Uganda, has suffered the negative effects of armed conflicts that negatively affect the provision of health care.”
There’s no reference here. Perhaps you could see if ACLED has any info on the number or trend of conflict events in that region? https://acleddata.com/#/dashboard
As for how it impacts health provision, are there any reports you could reference?
“Although the mortality rate in the PLHIV population has declined in the DRC, the chance of survival is not the same for all patients.” No supporting information.
“Although the number of AIDS-related deaths decreased by 60% in 2018 in the DRC, the difficult of accessing ARV in the eastern region as a result of armed conflicts indicates that deaths will again increase.” No reference here.
Your survival curves appear to cross one another. How did this impact the proportional odds assumption for your cox models?
For table 4 you need to clarify what it means to die “related to armed conflict.” This is not clear in the paper.
Minor issues:
“We report that one-fourth of those that died in the cohort were direct victims of armed conflict.” I am not seeing where this is reported in the tables. You might include conflict data in the Table 1?
“Patients who had less ART experience at the time of enrollment had a greater mortality risk than those who had never had ART.” “Patients who had less experience with ART at enrolment had a higher risk of death compared with those who were naïve to ART”
“less experience” needs to be clarified in the text. Of course you have “status of treatment” in yoru tables, but it is not entirely clear what this means in terms of the statements above. Please clarify.
Section 3.3 Please include hazard ratios and confidence intervals in the text to better clarify the results in Table 3 within your text.
Author Response
Review of Determinants of Survival of HIV Patients Receiving Dolutegravir in Bunia, Democratic Republic of Congo: A Prospective Cohort Study
This is a well done study of HIV outcomes in a city near a region of conflict in the DRC. The authors have clearly stated they research goals, their data collection procedures and analytic methods. They have presented their results in a concise and clear manner. This article will be of interest to researchers to those working on HIV related topics and conflict/health. I do not feel as there is much that should prevent publication of this manuscript in IJERPH, but have the following suggestions.
Authors’ response: thank you for your feedback, which has helped us improve our manuscript.
Major issues:
The authors make many statements in the introduction without providing evidence or supporting information.
Example:
“Particularly since 2017, the city of Bunia, located in the eastern part of the DRC in the vicinity of Uganda, has suffered the negative effects of armed conflicts that negatively affect the provision of health care.”
There’s no reference here. Perhaps you could see if ACLED has any info on the number or trend of conflict events in that region? https://acleddata.com/#/dashboard
As for how it impacts health provision, are there any reports you could reference?
“Although the mortality rate in the PLHIV population has declined in the DRC, the chance of survival is not the same for all patients.” No supporting information.
“Although the number of AIDS-related deaths decreased by 60% in 2018 in the DRC, the difficult of accessing ARV in the eastern region as a result of armed conflicts indicates that deaths will again increase.” No reference here.
Authors’ response: we have reworked the entire introduction, which now reads as follows:
“1. Introduction
Globally, AIDS-related deaths have decreased since the peak in 2004, from 1.9 mil-lion in 2004 to 680,000 in 2021 [1]. As a result, the life expectancy of people living with HIV (PLHIV) has considerably improved due to highly effective antiretroviral (ARV), and the death rate among PLHIV is nearing that of the general population [2].
In the Democratic Republic of Congo (DRC), AIDS-related deaths decreased by 60% in 2018 [3]. However, this reduction in deaths will likely be reversed due to disrupted access to antiretroviral therapy (ART) by contextual factors, such as the COVID-19 pandemic, Ebola outbreaks, and continuous insecurity threats. At the patient level, re-sistance to dolutegravir (DTG)-containing regimens may also affect mortality [4].
Particularly since 2017, armed conflicts have negatively affected the provision of healthcare services in Bunia, the capital of the eastern province of Ituri, neighboring Uganda. Bunia is surrounded by violent combat zones, where the army and militia fight on a daily basis. Intermittently, clashes also occur in Bunia itself. Restricted access to healthcare services due to armed violence has been documented worldwide, such as in South America, where armed conflicts in Colombia have led to the forced displacement of millions of families [5]. In such insecure environments, the affected community may avoid visiting health facilities for fear of violence, rape, or even murder by the insur-gents [6, 7]. As a result, patients on ART may face challenges in following their routine ART programs and care schedules and experience a rebound in viral load (VL) and sometimes death [8].
Although the mortality rate among PLHIV has declined in the DRC, the survival chance is not the same for all. Mortality predictors include advanced AIDS clinical stages, the presence of comorbidities and opportunistic infections, lower body mass index, male sex, and high viral load during the first two years of illness [9–13]. In re-source-limited and especially conflict-stricken settings, such as in the DRC, the factors contributing to the death of PLHIV on DTG-based regimens are not yet well-elucidated. Therefore, we conducted a prospective cohort study aimed at investigating the survival determinants of PLHIV receiving DTG-containing ART in conflict-affected Bunia.”
- Methods
Your survival curves appear to cross one another. How did this impact the proportional odds assumption for your cox models?
Authors’ response: XXXX Pierre
For table 4 you need to clarify what it means to die “related to armed conflict.” This is not clear in the paper.
Authors’ response: we have clarified the text explaining the table. It reads as follows:
“Among the 23 reported deaths, six (26.1%) were directly related to armed conflicts. Although not statistically significant, the proportion of deaths as a direct result of armed conflict was higher among women and among those who were at stage I or II. Patients who died directly from armed conflict were younger than their counterparts (Table 4).”
Minor issues:
“We report that one-fourth of those that died in the cohort were direct victims of armed conflict.” I am not seeing where this is reported in the tables. You might include conflict data in the Table 1?
Authors’ response: please see our answer to the above question:
“Among the 23 reported deaths, six (26.1%) were directly related to armed conflicts. Although not statistically significant, the proportion of deaths as a direct result of armed conflict was higher among women and among those who were at stage I or II. Patients who died directly from armed conflict were older than their counterparts (Table 4).”
“Patients who had less ART experience at the time of enrollment had a greater mortality risk than those who had never had ART.” “Patients who had less experience with ART at enrolment had a higher risk of death compared with those who were naïve to ART”
“less experience” needs to be clarified in the text. Of course you have “status of treatment” in your tables, but it is not entirely clear what this means in terms of the statements above. Please clarify.
Authors’ response: we have now clarified it throughout the manuscript. For example, in the results:
“Regarding baseline treatment status, 62.2% were already receiving ART (experienced patients), 19.9% were self-reported naïve with a baseline VL of <50 copies/mL (new but with VL suppressed), and 17.9% were self-reported naïve with a baseline VL of ≥50 copies/mL (new but with high VL) (Table 2).”
“ART-naïve participants with a baseline VL of <50 copies/mL (i.e., suppressed) had a higher risk of death than participants already on ART (aHR: 5.43; 95% CI: 1.59–18.48).”
Section 3.3 Please include hazard ratios and confidence intervals in the text to better clarify the results in Table 3 within your text.
Authors’ response: we have now added the HR and CI as follows:
“After adjustment, older participants had a higher risk of death than the reference group (<35 years) (adjusted hazard ratio (aHR) for the 35-45 age group: 5.14; 95% CI: 1.09–24.18; aHR for the >45 age group: 5.47; 95% CI: 1.13–26.38). ART-naïve participants with a baseline VL of <50 copies/mL (i.e., suppressed) had a higher risk of death than participants already on ART (aHR: 5.43; 95% CI: 1.59–18.48). Participants with a high baseline VL (≥1000 copies/mL) had a higher mortality risk than the reference group (aHR: 6.04; 95% CI: 1.78–20.43) (Table 3 and Figures 1-3).”
Round 2
Reviewer 1 Report
Major issues:
The authors have significantly revamped the manuscript to be at a level a reader can now focus on the results and message of this research. However, I still have serious concerns about the conclusion of the study. The study is a prospective, case-control investigation with mortality as primary outcome. Overall 23 patients died, of whom 6 were directly related to armed conflicts. The authors did not link the mortality results with the treatment. The readers would want see if there is excess mortality in index group that can be attributed to armed conflicts. Such analysis would determine the effect of the war on people living with HIV. Such analysis could also be done in the control population. The authors concluded that armed conflicts in the region has played a significant role in mortality, while there was not a statistically significant value for such assertion. Looking at the data presented in Table 4, clearly armed conflicts did not significantly influence mortality, or at least the number was not large enough to make a good comparative analysis to determine the effect of war. Conversely, the authors cannot even say that armed conflicts did not play a role due to small sample size. That being said, the authors should have focused on other parameters such as VL that significantly played a role in mortality rate.
Minor issues:
Abstract: the use of “not least” is not correct.
Introduction, 2nd paragraph: “At the patient level, resistance to dolutegravir (DTG)-containing regimens may also affect mortality.” This sentence comes from no where. It should be introduces.
3rd paragraph: “Particularly” not properly used and should be dropped.
Parag. 4: “male sex” is vague.
Page 11, Statistical analysis: “which we found to be insignificant - the mean VIF was 1.73”. This sentence should be in the result section; not here.
page 11: “were self-reported naive” should read “self reported as naive”.